# An Eccentricity Error Separation Method for Rotary Table Based on Phase Feature of Moiré Signal of Single Reading Head

Yao Huang [1,2,*], Shuangliang Che [1], Weibin Zhu [3,*], Cheng Ma [3], Yi Zhou [3], Wei Zou [2] and Zi Xue [2]

1 State Key Laboratory of Modern Optical Instrumentation, College of Optical Science and Engineering, Zhejiang University, Hangzhou 310027, China
2 National Institute of Metrology, China, Beijing 100029, China
3 School of Measurement and Testing Engineering, China Jiliang University, Hangzhou 310018, China
* Correspondence: huangyao@nim.ac.cn (Y.H.); zhuweibin@cjlu.edu.cn (W.Z.)

**Abstract:** In view of the limitations of the existing eccentricity error separation method of the rotary table, an eccentricity error separation method based on the phase feature of the moiré signal of a single reading head is proposed herein. A grating pair transmission model is established based on the analysis of the principle of the rotary table; thereby, the influence of eccentricity error on the phase feature of the moiré signal in the rotation course of the rotary table is clarified, and the corresponding model between the phase feature spectral components and the eccentricity error is established. The verification experiments of the proposed method are carried out based on the laboratory-made circuit system. After verifying the accuracy of the data acquisition of the laboratory-made circuit board, the verification experiments of the eccentricity error separation effect of the proposed method are carried out. The experimental data are compared with those of the traditional method, and the results show that the error between the two methods is 2.34 μm, while the relative error is 2.3%.

**Keywords:** rotary table; eccentricity error; moiré signal; phase feature

## 1. Introduction

The rotary table plays a crucial role in the field of high-precision angle measurement. They can be broadly classified as mechanical, electromagnetic, and photoelectric, with photoelectric rotary tables being the most widely used, offering the advantages of high accuracy [1], a wide measurement range, and high immunity to interference. The photoelectric rotary table consists of a rotating shaft, a photoelectric reading head, and an internal circular grating. During the rotation process, the relative movement of the grating pair produces a changing moiré fringe, and the moiré signal corresponds to the displacement output by the photoelectric reading head [2]. There are quite a few factors affecting the accuracy of the rotary table [3], which can be roughly divided into inherent error and application-dependent error [4], of which the eccentricity error in the application-dependent error is the most important factor affecting the accuracy of angle measurement [5–7]; therefore, separating the eccentricity error is of great value to ensure the accuracy of the rotary table.

The separation methods for the eccentricity error of the rotary table can be divided into two types [8–11]. One is based on the fundamental circumferential closure property, wherein external calibration devices are used to separate the eccentricity error components of the rotary table by calibrating the discrete positioning error curve of the entire circumference of the rotary table. Chen Xi-jun et al. [12] established the moiré fringe equation with eccentricity error, derived the equation of angle measurement error caused by eccentricity error, calibrated rotary table using a polyhedral prism, fitted the angle measurement error equation of the rotary table and the eccentricity error of the rotary table, and improved the measurement accuracy range from $(-7.0'', 1.1'')$ to $(-1.9'', 1.0'')$ with the error compensation. Mi Xiao-tao et al. [13] analyzed the relationship between the angular measurement results and the error caused by the eccentricity error, calibrated the

discrete angular measurement error using a polyhedral prism and autocollimator, fitted the angular measurement error function through the least squares fitting method, separated the eccentricity error of the circular grating, and significantly reduced the angle-measurement error range by correcting from ($-31.6''$, $41.7''$) to ($-1.0''$, $1.2''$). Zheng Da-teng, et al. [14] established the relationship model between the eccentricity error and the angle measurement error of AACMM (Articulated Arm Coordinate Measuring Machine), calibrated the angle measurement error with a polyhedral prism, fitted the compensation function of the angle measurement error and the eccentricity error with a nonlinear least squares method, and increased the measurement accuracy by 44.6% with correction of eccentricity errors.

The advantage of this kind of method is that it can ensure high accuracy in separating the eccentricity error, and its disadvantage is that during the separation process, due to the use of calibration instruments, such as a polyhedral prism and autocollimator, the requirements of the operating environment and the proficiency are high, and there are cumbersome steps in the operation and subsequent processing.

Another kind of method was realized with various sensors to achieve eccentricity error separation in the rotary table [15–17]. Ai Cheng-guang et al. [18] proposed an eccentricity error separation model based on non-diametric dual reading heads, compared the phase difference of the moiré signals received by the dual reading heads, separated the eccentricity error and the eccentricity error direction of the circular grating in the platform by synthesizing the Lissajous graph of the signals, corrected the testing system based on the derived compensation formula, and increased the circular angle accuracy by nearly five times. Feng Chao-peng et al. [19] developed a model for the eccentricity error of dual reading heads, derived a self-calibration equation for the eccentricity error parameters of a circular grating based on the dual reading heads, and solved for the eccentricity error parameters by experimentally self-calibrating the eccentricity error parameters of a circular grating with double reading heads installed diametrically opposite, thereby achieving the separation of the eccentricity error of a circular grating. The error of the circular grating decreased from $0.0464°$ before compensation to $0.0037°$ after compensation. A fast least squares fitting method for calculating eccentricity error was proposed by Zhu San-ying et al. [20], who combined a laser displacement sensor with the proposed fast algorithm to accurately separate the eccentricity error of the device 40–50% faster than the calculation of LSF (least squares fitting). Wang Ya-zhou et al. [21] proposed a grating eccentricity error detection system based on image-based angular displacement measurement and used two pairs of diameter image sensors to separate the grating eccentricity error, reducing the root mean square error from $1017''$ to $12.8''$.

While these kinds of methods reduce the complexity of separating eccentricity errors, they have the disadvantage of requiring multiple sensors to obtain information on the eccentricity error. During the operation process, it is usually necessary to expose the rotating shaft and grating disc, which increases the risks to the internal precision structure of the rotary table and reduces angle measurement accuracy.

In summary, the existing two kinds of eccentricity error separation methods either rely on external instrument equipment or require multiple sensors and exposed turntable structures. Focusing on the limitations of the current methods, a new method for separating eccentricity error is proposed in this paper. This method neither relies on external equipment nor exposes the internal structure of the rotary table, and can separate the eccentricity error only using a single reading head of the rotary table. By establishing a grating pair transmission model, the influence of eccentricity error on the phase feature of the moiré signal in the rotation course of the rotary table is clarified; thereby, a model for the relationship between the phase spectrum components and eccentricity error is established. The verification and comparison experiments of the eccentricity error separation method are carried out based on the laboratory-made circuit system, and the experimental results prove the effectiveness and accuracy of the proposed method. The new method has the advantages of low cost, simple steps, low environmental requirements, and little impact on the rotary table.

In the first part of the introduction, the background and significance of the research on eccentricity error separation are introduced, the classification and related achievements of eccentricity error separation methods for the rotary table are described in detail, and the innovation and research content of this paper are clearly and accordingly expressed. In the second part, the principle of the photoelectric rotary table is explained first. The grating transmission model and displacement error model caused by eccentricity error are established. The relationship between eccentricity error and the phase characteristics of the moiré signal is demonstrated, and the correctness and feasibility of the eccentricity error separation method proposed in this paper are proved. In the third part, the functional effectiveness of the experiment of the laboratory-made circuit board for collecting moiré signals is proved first. The experimental system and equipment are described in detail. By comparing the results of eccentricity error obtained by the proposed method in this paper with those of external calibration instruments, the accuracy and effectiveness of the method proposed in this paper are proved. In the first part of the conclusion, the full text is summarized, and the promotion value and application occasions of the proposed method are demonstrated.

## 2. Model of the Proposed Method

The photoelectric rotary table is mainly composed of a rotating shaft, an internal circular grating, and a grating reading head. The internal grating in the reading head is the indicative grating, and the circular grating is the scale grating. The two of them form the grating pair. The schematic of the structure of the rotary table is shown in Figure 1.

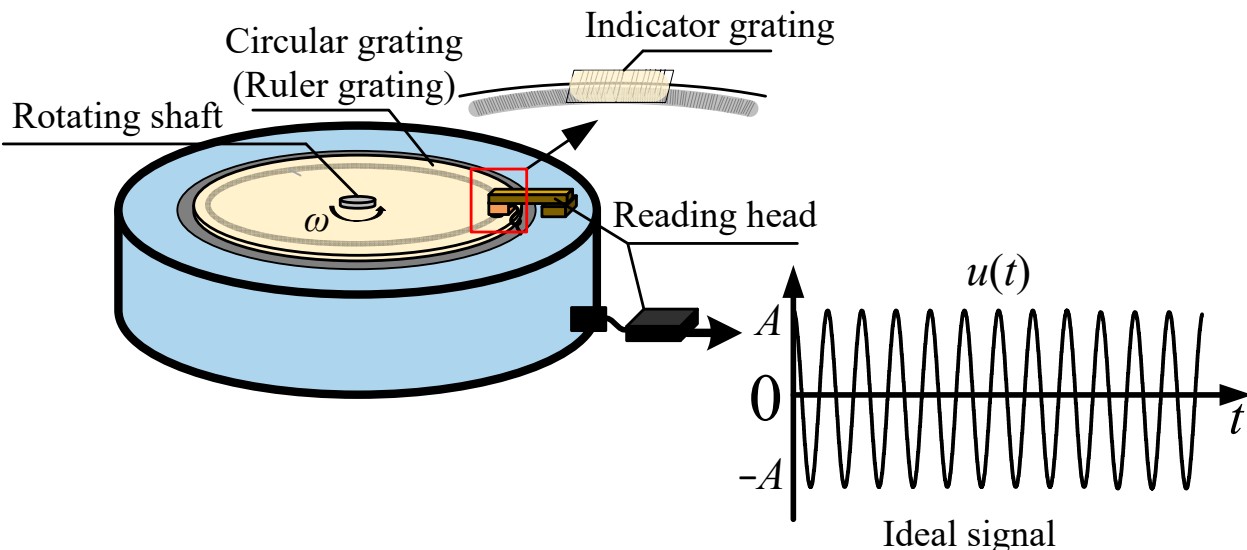

**Figure 1.** Schematic of structure of the rotary table.

As shown in Figure 1, when rotating at a constant speed $\omega$, the output moiré signal, defined as $u(t)$, has a constant frequency and amplitude. The period of $u(t)$ corresponds to the indexing angle of the grating line.

Due to the small receiving field of the reading head, usually at the millimeter level, the indicative grating and scale grating of the rotary table can both be regarded as a single rectangular grating, as shown in Figure 2.

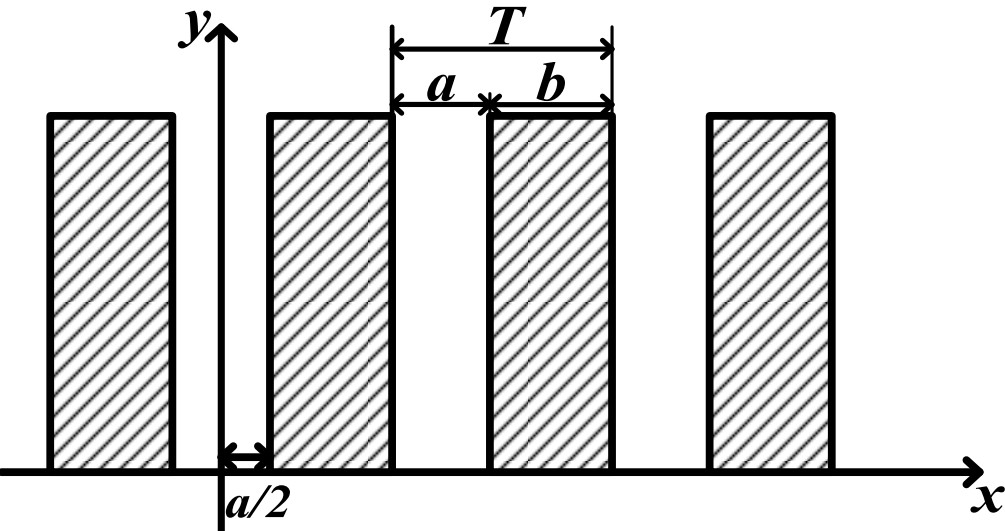

**Figure 2.** Schematic of a single rectangular grating.

In the *xoy* plane where the grating lines are located, the *x* direction is the vertical grating line direction, and the grating lines are evenly distributed. Set the grating pitch is $T$, the length of the shading area is $b$ with the light transmittance is zero, and the length of the light-transmitting area is $a$ with the light transmittance is equal to one. For the parallel light emitted from the reading head, grating transmission function $L(x)$ can be described as:

$$L(x) = \begin{cases} 1 & \left[ kT - \dfrac{a}{2}, kT + \dfrac{a}{2} \right] \\ 0 & \text{else} \end{cases} \tag{1}$$

Extend the Fourier Series on $L(x)$, and its complex form can be written as:

$$L(x) = \sum_{n=-\infty}^{\infty} A_n \exp(i2\pi nvx) \tag{2}$$

where $v$ is the spatial frequency of the grating and $v = 1/T$, $A_n$ is the Fourier coefficient and $n = 0, \pm 1, \pm 2$, etc. The expression of $A_n$ is shown in Equation (3).

$$\begin{aligned} A_n &= \frac{1}{T} \int_0^d L_1(x) \exp\left( \frac{-i2\pi nx}{T} \right) dx \\ &= \frac{1}{\pi n} \sin\left( \frac{\pi na}{T} \right) \\ &= \frac{1}{\pi n} \sin(\pi n \beta) \end{aligned} \tag{3}$$

where $\beta = a/T$. The transmission functions of the scale grating and the indicative grating follow the above basic equation.

In the actual measurement process, the indicative grating and the scale grating do not completely overlap, and there is a certain angle of $\gamma$. The schematic diagram of the relative position of the two gratings is shown in Figure 3.

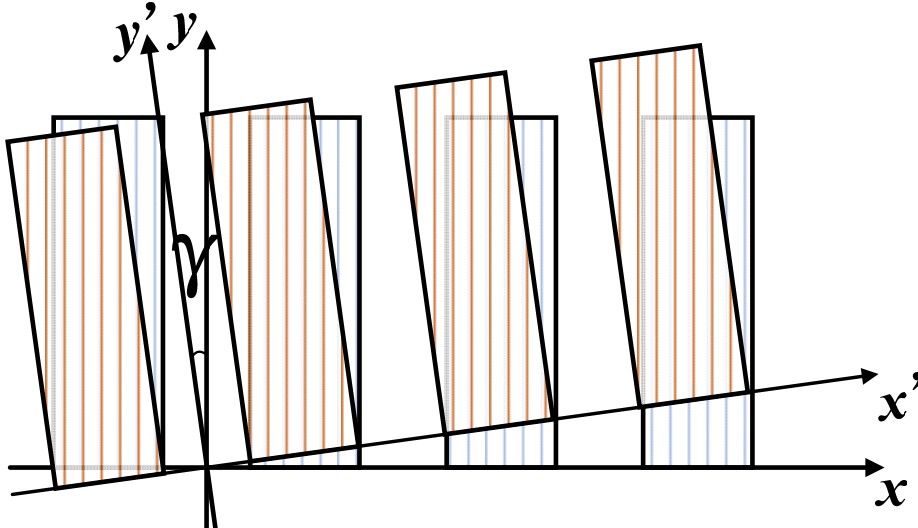

**Figure 3.** Schematic of relative position of indicative grating and scale grating.

When the indicative grating and the scale grating are superimposed, the indicative grating decomposes the incident light into plane wave arrays in different propagation directions, namely expanding the incident light into the amplitude distribution of the cosine primitive. For the scale grating, the plane waves in each direction of the indicative grating are incident waves, that is to say, the indicative grating decomposes the parallel light and the scale grating subsequently modulates the decomposed light field. The transmission functions of $L_1(x)$ and $L_2(x, y)$ of the two gratings can be expressed as:

$$\begin{cases} L_1(x) = \sum\limits_{n=-\infty}^{\infty} A_n \exp(i2\pi n v_1 x) \\ L_2(x, y) = \sum\limits_{m=-\infty}^{\infty} B_m \exp[i2\pi m v_2 (x \cos \gamma + y \sin \gamma)] \end{cases} \tag{4}$$

where $v_1$ and $v_2$ are the spatial frequencies of the indicative grating and scale grating, respectively. $A_n$ and $B_m$ are the Fourier Series of the two transmission functions, respectively. The transmission function $M(x, y)$ of the moiré fringe composed of the scale grating and the indicative grating is:

$$\begin{aligned} M(x, y) &= L_1(x) \cdot L_2(x, y) \\ &= \sum\limits_{n=-\infty}^{\infty} A_n \exp(i2\pi n v_1 x) \cdot \sum\limits_{m=-\infty}^{\infty} B_m \exp[i2\pi m v_2 (x \cos \gamma + y \sin \gamma)] \end{aligned} \tag{5}$$

Equation (5) is the transmission function of the grating pair inside the rotary table under ideal conditions.

In an actual working situation, the rotary table is affected by factors such as mechanical wear and installation errors, and thus the internal attitude will deviate. The actual structure of the internal section of the rotary table and the output moiré signal, which is defined as $u'(t)$, is shown in Figure 4.

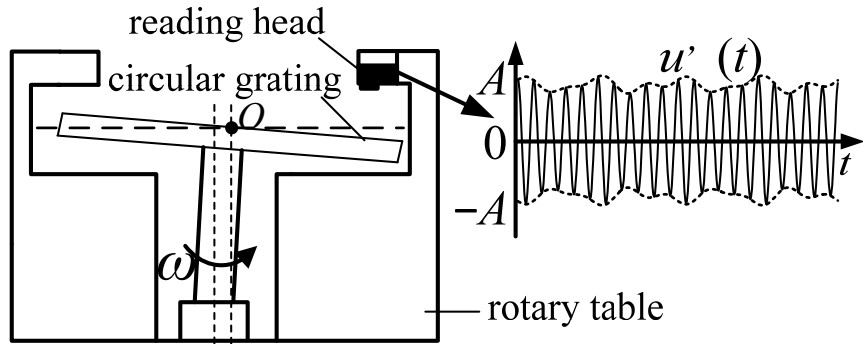

**Figure 4.** Schematic of the rotary table in actual working condition.

It can be seen from Figure 4 that due to the change in the internal attitude of the rotary table, the relative position of the indicative grating and the scale grating is displaced. The output light intensity of the incident light irradiated in the grating pair changes, and subsequently, the output moiré signal is different from the ideal moiré signal. The difference in amplitude and phase between the output moiré signal and the ideal moiré signal is collectively called the feature of moiré signal.

The phase feature of the moiré signal, defined as $\varphi_m(t)$, is used to realize the eccentricity error separation of the rotary table in this paper. Since the rotary table is affected by quite a few factors in the actual rotation, the $\varphi_m(t)$ can be regarded as a composite waveform formed by the superposition and mixture of various frequency harmonics, as shown in Figure 5.

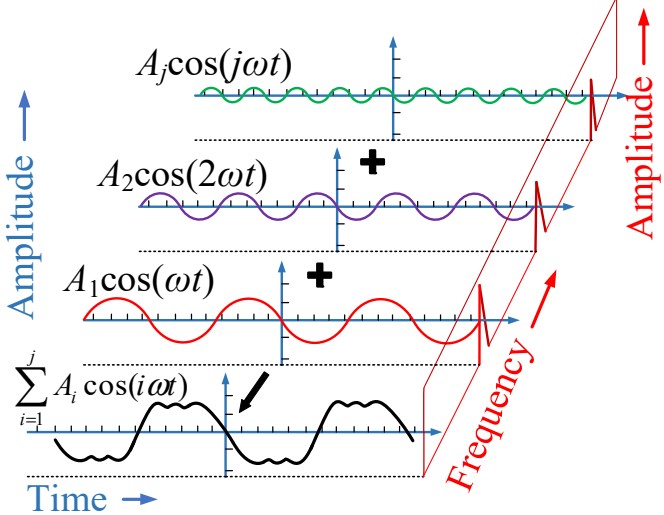

**Figure 5.** Schematic of $\varphi_m(t)$.

In this figure, $A_i\cos(i\omega t)$, $i = 1$, 2, etc., are the harmonic components of the feature of the moiré signal caused by system error factors of the rotary table, where $A_i$ is the amplitude of the harmonic and $i$ is the order of the harmonic. The feature of moiré signal of the rotary table is formed by the superposition and addition of harmonic components of each order. Since the period of the eccentricity error of the rotary table is $2\pi$, the phase feature of the moiré signal caused by eccentricity error, defined as $\varphi_{me}(t)$, corresponds to the first-order component of $\varphi_m(t)$, which accounts for the largest proportion in the feature of moiré signal, and has the greatest influence on the amplitude and phase of the moiré signal.

The schematic diagram of the displacement error of the rotary table affected by the eccentricity error is shown in Figure 6.

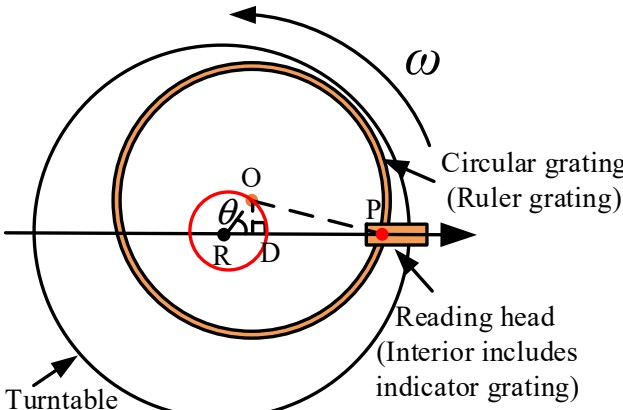

**Figure 6.** Schematic of the displacement error caused by the eccentricity error of the rotary table.

As shown in Figure 6, when the rotary table rotates at the speed of $\omega$, due to the eccentricity error, defined as $e$, the grating rotation center, R, and the geometric center, O, do not coincide. A displacement error, defined as $L_e(\theta)$, will be generated during the rotation of the grating with the rotary table.

Establish the $x$-axis from the grating rotation center, R, to the reading head in the positive direction. The intersection point between the outer side of the marking line of the circular grating and the positive semi-axis of the $x$-axis is defined as $P$. Subsequently, $OP$ is the radius of the grating, and the variation of $RP$ with the angular position of the rotary table, $\theta$, is defined as $L_e(\theta)$, namely, the displacement error of the rotary table.

Since $OD \perp RP$, as well as $RO$ and $RD$, are at a micro level in actual working conditions, it is thus clear that:

$$\cos \angle OPD = \frac{DP}{OP} \approx 1 \tag{6}$$

The $L_e(\theta)$ can be described as:

$$L_e(\theta) = RP - OP \approx RD = |RO| \cos \theta = e \cdot \cos \theta \tag{7}$$

$L_e(\theta)$, which is introduced by the eccentricity error of the rotary table, will affect the transmission function of the grating pair of the rotary table. The transmission function of the scale grating affected by the eccentricity error, defined as $L_2(x - x'(\theta), y)$, can be expressed as:

$$L_2(x - x'(\theta), y) = \sum_{m=-\infty}^{\infty} B_m \exp\left\{ i2\pi m v_2 \left[ (x - x'(\theta)) \cos \gamma + y \sin \gamma \right] \right\} \tag{8}$$

where $x'(\theta)$ is the relative displacement of the two gratings, which is changing with $\theta$, caused by the attitude error of the rotary table. At this time, the transmission function of the moiré fringe composed of the scale grating and the indicative grating, $M(x, y)$, can be described as:

$$
\begin{aligned}
M(x, y) &= L_1(x) \cdot L_2(x - x'(\theta), y) \\
&= \sum_{n=-\infty}^{\infty} A_n \exp(i2\pi n v_1 x) \cdot \sum_{m=-\infty}^{\infty} B_m \exp[i2\pi m v_2((x - x'(\theta)) \cos \gamma + y \sin \gamma)]
\end{aligned} \tag{9}
$$

The moiré fringe consists of contents produced by the beat phenomenon between two gratings. The content of the largest spatial period in the light intensity distribution constitutes the fundamental wave of the moiré fringe, and the general expression of the largest period moiré fringe produced by the beat phenomenon is obtained by taking $m = -n$ in Equation (9).

$$
\begin{cases}
V_x = n v_1 - n v_2 \cos \gamma \\
V_y = -n v_2 \sin \gamma
\end{cases} \tag{10}
$$

In Equation (10), $V_x$ and $V_y$ are the components of the spatial frequency of the light intensity function in the *x*-axis and *y*-axis directions, respectively, and both of them are constants. Defining the intensity of the parallel light emitted from the reading head as $I_r$, the outgoing light intensity generated by the light passing through the indicative grating and scale grating, defined as $I_c(x - x'(\theta), y)$, can be expressed as:

$$
\begin{aligned}
I_c(x - x'(\theta), y)_{m=-n} &= I_r \cdot M(x, y) \\
&= I_r \beta_1 \beta_2 + 2I_r \sum_{n=1}^{\infty} A_n B_{-n} \cos\left\{ 2\pi \left[ (xV_x + yV_y) + v_2 x'(\theta) \cos \gamma \right] \right\}
\end{aligned}
\tag{11}
$$

When the photoelectric conversion rate of the photoelectric receiver is *k* and the rotational speed of the rotary table is $\omega$, the output moiré signal by the rotary table, $u'(t)$, can be expressed as:

$$
\begin{aligned}
u'(t) &= I_c(x - x'(\omega \cdot t), y)_{m=-n} \cdot k \\
&= kI_r \beta_1 \beta_2 + 2kI_r \sum_{n=1}^{\infty} A_n B_{-n} \cos\left\{ 2\pi \left[ (xV_x + yV_y) + v_2 x'(\omega \cdot t) \cos \gamma \right] \right\}
\end{aligned}
\tag{12}
$$

From Equation (12), it can be seen that the rotary table is affected by the eccentricity error during the rotation process, and the introduced phase feature of the moiré signal, $\varphi_{me}(t)$, can be expressed as:

$$
\phi_{me}(t) = 2\pi \cdot v_2 \cdot e \cdot \cos(\omega \cdot t) \cdot \cos \gamma
\tag{13}
$$

When the grating pitch is 20 μm, the spatial frequency $v_2$ of the scale grating in the rotary table is $1/20$ μm. In an actual working condition, the angle of $\gamma$ between the indicative grating and the scale grating in the rotary table is insignificant, and the $\cos\gamma$ in Equation (13) is approximately equal to one, so Equation (13) can be simplified as:

$$
\phi_{me}(t) = 2\pi \cdot v_2 \cdot e \cdot \cos(\omega \cdot t)
\tag{14}
$$

Since $\varphi_{me}(t)$ is the first-order component of $\varphi_m(t)$ in the frequency domain, and the corresponding frequency-domain amplitude is $A(1)$, according to Equation (14), the relationship between the first-order amplitude, $A(1)$, and the eccentricity error, e, is as follows:

$$
A(1) = (\phi_{me}(t))_{\max} = 2\pi \cdot v_2 \cdot e
\tag{15}
$$

The expression of the eccentricity error of the rotary table is:

$$
e = \frac{A(1)}{2\pi \cdot v_2}
\tag{16}
$$

Since $v_2$ is the spatial frequency of the scale grating, which is a constant in a fixed rotary table, it can be seen that the eccentricity error of the rotary table is directly proportional to the first-order component of $\varphi_m(t)$ in the frequency domain.

### 3. Results and Discussion

*3.1. Verification of Moiré Signal Data Acquisition*

Since the separation method of the eccentricity error is based on the moiré signal of the rotary table, the quality of the moiré signal collected by the hardware circuit is crucial to the separation result of eccentricity error. In this paper, the acquisition of a full-circle moiré signal of the rotary table is realized by using a laboratory-made circuit board with FPGA, named GSA_B, which is shown in Figure 7.

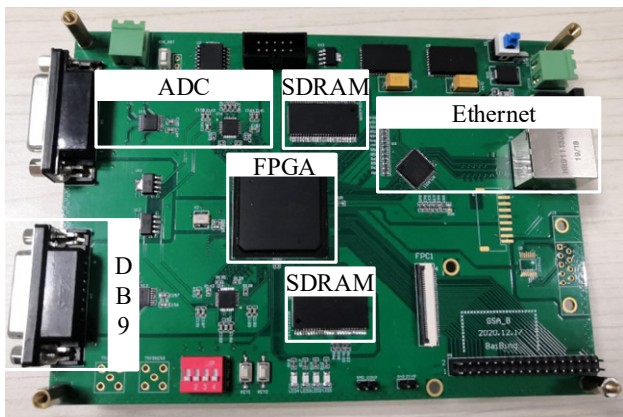

**Figure 7.** Laboratory-made circuit board named GSA_B.

As shown in Figure 7, GSA_B uses FPGA as the main chip, and the analog signal output by the reading head of the rotary table is connected through the DB9 interface and converted into digital data through high-speed ADC; then, it enters the FPGA to process. The data storage module realizes the data exchange between FPGA and SDRAM, and the processed full-circle moiré signal is transmitted to the upper machine for analysis using Ethernet.

The effectiveness and correctness of the acquisition of moiré signals are verified with GSA_B. The moiré signal of a single reading head of the rotary table in the rotation course of the entire circumference is collected. FFT analysis is performed on the collected moiré signal, and the hardware acquisition function is verified by checking whether the main frequency of the signal matches the rotational speed of the rotary table.

The rotation rate of the rotary table is set to $120°/s$, and the time domain diagram and frequency domain diagram of the full-circle moiré signal, as shown in Figure 8.

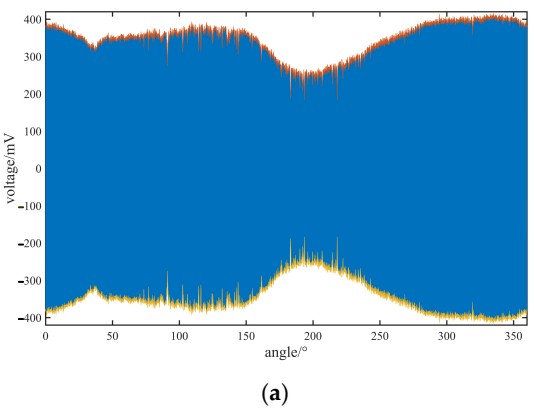

(**a**)

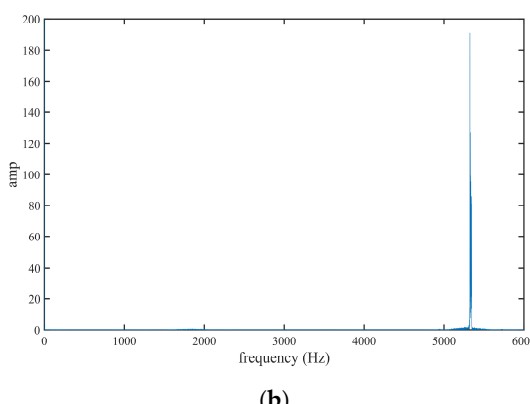

(**b**)

**Figure 8.** Schematic of moiré signal waveform of the whole circle. (**a**) The time domain diagram of collected signal. (**b**) The frequency domain diagram of collected signal.

The collected whole circle discrete moiré signal of single reading head by GSA_B is defined as $u(\theta)$. The displayed waveform in Figure 8a removes the DC amount of the moiré signal and adds the data envelopment analysis. It can be seen that during the full circle rotation of the rotary table, the upper and lower envelopes of the moiré signal are smooth, and there is no loss or jump. Affected by various errors during the rotation of the rotary table, the moiré signal envelope presents fluctuations. The changes of amplitude and phase of $u(\theta)$ in each cycle prove the existence of the feature of the moiré signal of the rotary table.

The schematic of the frequency domain of the collected moiré signal after FFT transformation is shown in Figure 8b. The relationship between the moiré signal frequency, $f$, and the rotary table speed, $\omega$, can be expressed as:

$$f = \frac{1}{\dfrac{360}{K \cdot \omega}} = \frac{\omega \cdot K}{360} \tag{17}$$

where $K$ is the number of circumferential lines on the grating disc and $K = 16{,}384$ in this experiment.

Substituting the set speed $\omega = 120°/\text{s}$ of the rotary table into Equation (17), the reference frequency can be obtained as 5461 Hz. It can be seen in Figure 8b that the main frequency of the full-circle moiré signal collected is around 5325 Hz. The relative error with the reference value is about 2.5%. Since the rotary table cannot guarantee a completely uniform speed during the entire circle rotation process, it can be considered that the collected data are basically consistent with the set speed.

### 3.2. Validation of the Effect of Eccentricity Error Separation

In order to verify the feasibility and effectiveness of the separation method proposed in this paper for the eccentricity error, experiments with the proposed method and the traditional method using calibration devices were carried out on the same rotary table, respectively, and the experimental results were compared.

#### 3.2.1. Experiment of Separation Method with Calibrated Instrument

The traditional method of separating the eccentricity error of the rotary table by calibration devices is realized by using the autocollimator combined with the polyhedral prism to obtain the discrete numerical sequence of the positioning error of the rotary table at equal intervals. The eccentricity error of the rotary table is obtained by fitting the positioning error points. The main devices and their parameters used in the experiment are shown in Table 1.

**Table 1.** Specifications of main instruments.

| Instrument | Model (Manufacturer) | Specification |
|:---:|:---:|:---:|
| Autocollimator | ELCOMAT-3000 (MOLLER-WEDEL OPTICAL) | From $-1000''$ to $+1000''$, MPE: $0.1''$ |
| Polyhedral prism | 24-sided secondary polyhedral prism | Grade 2 |
| Rotary table | Air-bearing rotary table | Repeatability: $0.3''$, Accuracy $\pm 0.5''$ |
| Grating disc | R10851 (MicroE system) | Grating line number: 16,384, Grating pitch: 20 μm, Radius: 53.975 mm, |
| Reading head | M1000 (MicroE system) | Rotary: up to $\pm 2.1''$ |

The experimental setup for calibration using a polyhedral prism and an autocollimator is shown in Figure 9.

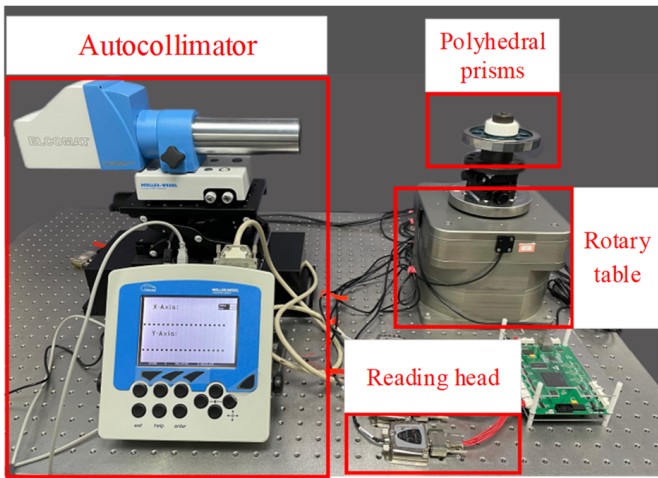

**Figure 9.** Photograph of experimental setup for calibration.

The entire circumference of the discrete positioning error array, *P*, obtained by the calibration setup, is shown in Table 2.

**Table 2.** Discrete positioning error on the entire circumference.

| Angle Value/° | Angle Measuring Error/″ | Angle Value/° | Angle Measuring Error/″ |
|---|---|---|---|
| 15 | 93.81 | 195 | −149.17 |
| 30 | 181.17 | 210 | −245.19 |
| 45 | 251.37 | 225 | −322.77 |
| 60 | 302.98 | 240 | −379.90 |
| 75 | 332.05 | 255 | −409.91 |
| 90 | 343.87 | 270 | −415.50 |
| 105 | 335.13 | 285 | −394.78 |
| 120 | 307.11 | 300 | −347.20 |
| 135 | 264.42 | 315 | −279.68 |
| 150 | 189.95 | 330 | −193.81 |
| 165 | 84.61 | 345 | −99.33 |
| 180 | −35.76 | 360 | −0.05 |

As shown in Table 2, within the entire circumference of the rotary table rotation, the positioning error of the rotary table, *P*, is unimodal. The maximum positioning error, $P_{\text{max}}$, and the minimum positioning error, $P_{\text{min}}$, only appear once. The equation for calculating the eccentricity error of the rotary table according to the peak value of *P* is:

$$e = \left( \frac{r \cdot \sin(P_{\text{max}})}{\sin(\theta_1)} + \frac{r \cdot \sin(P_{\text{min}})}{\sin(\theta_2)} \right) / 2 \qquad (18)$$

where $\theta_1$ is the corresponding rotary table rotation angle when the positioning error is $P_{\text{max}}$, while $\theta_2$ is the rotation angle corresponding to $P_{\text{min}}$. *r* is the radius of the grating disc inside the rotary table, and *r* = 53.98 mm in this experiment.

According to Table 2, $P_{\text{max}} = 343.87''$, $\theta_1 = 90°$, $P_{\text{min}} = -415.50''$, and $\theta_2 = 270°$. According to Equation (18), the value of the eccentricity error of the rotary table is 99.36 μm.

For the angle calibration value of each discrete position, the uncertainty of each contributor is shown in Table 3.

**Table 3.** The uncertainty of each contributor.

| No. | Contributor | Value |
|---|---|---|
| 1 | Indication error of autocollimator | 0.1″ |
| 2 | Uncertainty of polyhedral prism | 1.0″ |
| 3 | Uncertainty of installation error | 0.03″ |
| 4 | Uncertainty of repeatability | 0.19″ |

The combined uncertainty of the angle measuring error can be expressed as:

$$u_c = \sqrt{u_1^2 + u_2^2 + u_3^2 + u_4^2} \approx 1.02'' \tag{19}$$

Combining Equations (18) and (19), the uncertainty of the eccentricity error can be obtained as:

$$u = \sqrt{\left(\frac{\partial e}{\partial P_{\max}}\right)^2 \cdot u_c^2 + \left(\frac{\partial e}{\partial P_{\min}}\right)^2 \cdot u_c^2} = \sqrt{\left[\frac{r}{2 \cdot \sin(\theta_1)}\right]^2 \cdot u_c^2 + \left[\frac{r}{2 \cdot \sin(\theta_2)}\right]^2 \cdot u_c^2} = 0.01 \text{mm} \tag{20}$$

As the result is normally distributed, and the coverage probability is 95%, the coverage factor $k = 2$ and the expanded uncertainty can be calculated as:

$$u_g = k \cdot u = 0.02 \text{mm} \tag{21}$$

3.2.2. Experiment of the Eccentricity Error Separation with Proposed Method

The experiment is performed using the same setup shown in Figure 9. The discrete full-circle moiré signal, $u(\theta)$, is collected. The phase of the ideal moiré signal is used as a reference to obtain the phase feature of $u(\theta)$, defined as $\varphi(\theta)$, through the phase difference processing of each sampling point, as shown in Figure 10.

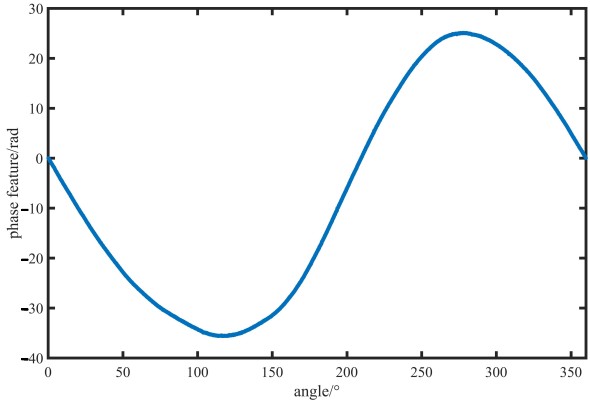

**Figure 10.** Schematic of the phase feature of $u(\theta)$.

It can be seen from Figure 10 that the phase error of the rotary table within the entire circumference is also unimodal. The phase difference fluctuates in the range of [−35.62 rad, 25.12 rad], which proves that the phase feature of the moiré signal is affected by the eccentricity error.

The FFT analysis is performed on $\varphi(\theta)$, and the transformation results in the frequency domain are shown in Figure 11.

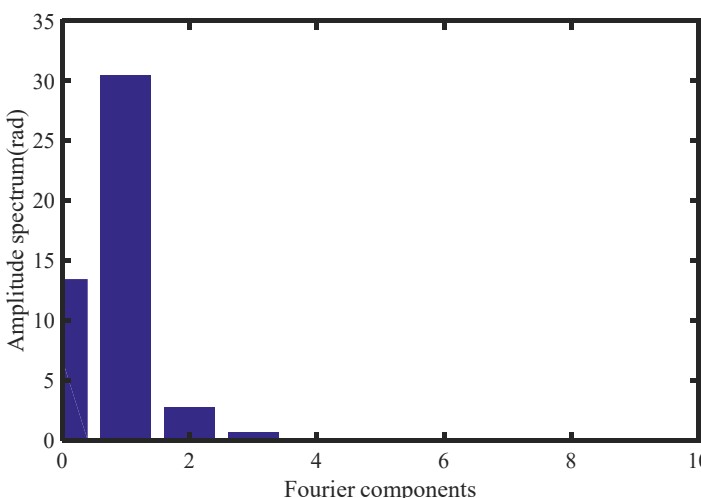

**Figure 11.** Results of spectrum analysis of $\varphi(\theta)$.

As stated in Figure 5, the phase feature of the moiré signal is composed of various frequency harmonics. The frequency component introduced by the eccentricity error during the rotation process is in accordance with the first-order frequency. It can be seen from Figure 11 that the first-order frequency component of $\varphi(\theta)$, $A(1)$, is 30.48 rad.

Based on the relationship between $A(1)$ and the eccentricity error expressed in Equation (16), and referring to the value of the grating pitch in Table 1, the eccentricity error can be calculated; the result is 97.02 μm.

As the values of the eccentricity error separated by the two methods are 99.36 μm and 97.02 μm, respectively, it can be considered that the eccentricity error value separated using the proposed method in this paper is consistent with that of the traditional method using calibration devices. The difference between the two algorithms is 2.34 μm, and the relative error is 2.3%. So, this proposed method has the feasibility and data accuracy in the actual rotary table application.

## 4. Conclusions

A new method for separating the eccentricity error of the rotary table based on the moiré signal of the single reading head is proposed in this paper. By establishing a grating pair transmission model, the effect of the eccentricity error on the phase feature of the moiré signal is clarified, and then an eccentricity error separation model based on the phase spectrum components is established. The validity and accuracy of the proposed method are demonstrated through the verification and comparison experiments. The value of the eccentricity error separated using the proposed method is compared with those of the traditional method with an autocollimator and polyhedral prism. The comparing results show that the difference is 2.34 μm, and the relative error is 2.3%. Due to the proposed method only relying on the signal of the rotary table to separate the eccentricity error, it can be achieved by simply obtaining the signal from the original reading head of the rotary table during the application process. The implementation process is simple, and the accuracy is relatively high, so it has obvious advantages in the field of in situ measurement. The proposed method provides a new method and new ideas for the detection of the working condition of the rotary table.

**Author Contributions:** Conceptualization, Y.H.; methodology, S.C.; software, C.M. and Y.Z.; validation, W.Z. (Weibin Zhu) and C.M.; formal analysis, Y.H., S.C. and W.Z. (Weibin Zhu); investigation, Y.Z. and W.Z. (Weibin Zhu); resources, Y.H. and W.Z (Wei Zou); data curation, S.C.; writing—original draft preparation, Y.H.; writing—review and editing, W.Z. (Weibin Zhu); visualization, Y.Z. and W.Z. (Weibin Zhu); supervision, Z.X.; project administration, W.Z. (Wei Zou); funding acquisition, Y.H. and W.Z. (Weibin Zhu). All authors have read and agreed to the published version of the manuscript.

**Funding:** This research was funded by the National Natural Science Foundation of China (grant number 52175526), the Research Project of General Administration of Quality Supervision, Inspection and Quarantine of PR China (2016QK189).

**Institutional Review Board Statement:** Not applicable.

**Informed Consent Statement:** Not applicable.

**Data Availability Statement:** The data underlying the results presented in this study are not currently publicly available but may be obtained from the authors upon request.

**Conflicts of Interest:** The authors declare no conflict of interest.

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
