# Peer review of "An Eccentricity Error Separation Method for Rotary Table Based on Phase Feature of Moiré Signal of Single Reading Head"

_photonics, doi:10.3390/photonics10111267_

Round 1

Reviewer 1 Report

Comments and Suggestions for Authors

In the manuscript “An Eccentricity Error Separation Method for Rotary Table Based on Phase Feature of Moiré Signal of Single Reading Head”, the authors proposed a method for separating the eccentricity error of rotary table, the final results are quite impressive. However, there are still some issues required to be addressed. The questions are as follows:

1. It is noticed that in 3.2.1, the calibration range is 180° to 360°, why only half a rotation cycle was calibrated? Please add the explanations.

2. It is suggested adding the calibration accuracy analysis of eccentricity error by the autocollimator in the manuscript, making the comparison results more convincing.

3. The advantages of the new method are suggested to be highlighted compared with the other existing methods.

Comments on the Quality of English Language

1.There are some spelling or formatting issues, and extra spaces exist between some words. It is suggested to promote a more detailed examination of the manuscript. 

Reviewer 2 Report

Comments and Suggestions for Authors

An eccentricity error separation method based on the phase feature of the moiré signal of single reading head is proposed to measure the rotational angle of the rotary table.

1. Structure and Organization: I recommend including a section in the introduction that outlines the structure of your paper. This will help readers gain a clear understanding of the logical flow of your research.

2. Background and Related Work: Provide more detailed information when introducing the problem background and related research. Analyze the strengths and weaknesses of each researcher's work, rather than giving a general overview.

3. Advantages and Innovation: Clearly highlight the advantages and innovation of your approach. How does it differ from existing methods? It would be helpful to compare the precision of your results with those achieved by other researchers.

4. Practical Applications: Discuss the practical applications of your method. Currently, there is no mention of how this approach can be applied in real system.

Reviewer 3 Report

Comments and Suggestions for Authors

The manuscript focuses on precision angle measurement using a photoelectric rotary table. The basic scientific idea is to separate eccentricity errors based on the phase feature of the moire signal of a single read head. Experimental testing on a laboratory installation confirmed the effectiveness of the proposed method.

These research results are original and have scientific value. Nevertheless, there are certain moments demanding explanations.

1. When analyzing the state of research, it is advisable to add information about devaces for measuring roundness, for example, O.V. Zakharov, and A.V. Kochetkov. Minimization of the systematic error in centerless measurement of the roundness of parts. Measurement Techniques, 2016. Vol. 58, pp. 1317-1321.

2. In Fig. Figure 4 shows that a circular grating has eccentricity and inclination. However, in further calculations the possible tilt of the circular grating is not taken into account.

3. Judging by Fig. 6 eccentricity is constant. Is this really true?

4. Results in Fig. 11 require some explanation.

5. The conclusion must have quantitative results.

Round 2

Reviewer 2 Report

Comments and Suggestions for Authors

The response is well. No further comments.